# De-Escalation of Therapy for Patients with Early-Stage Squamous Cell Carcinoma of the Anus

**DOI:** 10.3390/cancers13092099

**Published:** 2021-04-27

**Authors:** Eric Miller, Jose Bazan

**Affiliations:** Department of Radiation Oncology at the Arthur G. James Cancer Hospital and Richard J. Solove Research Institute, The Ohio State University Comprehensive Cancer Center, Columbus, OH 43210, USA; eric.miller@osumc.edu

**Keywords:** anal cancer, radiation therapy, chemoradiation, de-escalation of therapy

## Abstract

**Simple Summary:**

Management of early-stage squamous cell carcinoma of the anus (SCCA) remains controversial. The current standard of care treatment of chemotherapy combined with radiation therapy can result in both acute and late toxicity. Alternative therapies, including radiation therapy alone or local excision, may be less toxic, but the role of these therapies in early-stage SCCA remains unclear. Additional options for reducing the intensity of therapy for early-stage SCCA include reduction of radiation dose, altering treatment volumes, modifying chemotherapy type and dosage, and using intensity-modulated radiation therapy to reduce the radiation dose to adjacent normal tissues. Multiple prospective studies are actively investigating the role of de-escalation of therapy in patients with early-stage SCCA.

**Abstract:**

The incidence of squamous cell carcinoma of the anus (SCCA) is increasing, particularly in the elderly, with increased mortality in this age group. While the current standard of care for localized SCCA remains chemoradiation (CRT), completion of this treatment can be challenging with risks for severe acute and late toxicity. It remains unclear if full course CRT is required for the management of early-stage SCCA or if de-escalation of treatment is possible without compromising patient outcomes. Alternative therapies include radiation therapy alone or local excision for appropriate patients. Modifying standard CRT may also reduce toxicity including the routine use of intensity-modulated radiation therapy for treatment delivery, modification of treatment volumes, and selection and dosing of concurrent systemic therapy agents. Finally, we provide an overview of currently accruing prospective trials focused on defining the role of de-escalation of therapy in patients with early-stage SCCA.

## 1. Introduction

Squamous cell carcinoma of the anus (SCCA) remains a relatively rare malignancy, representing less than 1% of all cancer cases in the United States [1]. However, the incidence of SCCA is increasing, particularly in the elderly, with an almost 5% increase in mortality due to this malignancy in the most vulnerable [2]. While the current standard of care treatment for localized SCCA is radiation therapy (RT) with concurrent multiagent chemotherapy, questions remain if this potentially toxic and morbid treatment is appropriate for those with early-stage disease (e.g., T1–2 N0) [3,4,5,6]. In other HPV-related malignancies, such as head and neck cancer, there has been a recent emphasis placed on careful de-escalation of therapy to potentially mitigate against acute and late treatment-related toxicity while not adversely impacting patient outcomes [7]. Similarly, in SCCA, any adjustments to the currently accepted standard of care treatment that may potentially improve tolerability and reduce the risk of toxicity while delivering the same level of cancer control are welcome. This is of particular importance in early-stage SCCA, where rates of overall survival (OS) and disease-free survival (DFS) at 5 years are anticipated to be 86% and 80%, respectively, following treatment for patients with T2 N0 disease based on the results of RTOG-9811 [8]. Further, progress in imaging, resulting in better detection of metastatic or locally advanced disease [9,10], adds an additional layer of complexity in interpreting the results of earlier studies and evaluating treatment of patients considered to have early-stage disease. In this review, we will summarize the available literature focused on de-escalation of therapy for early-stage SCCA, including modifications in treatment modality as well as alterations in the currently accepted standard of care. We will also discuss modifications to standard chemoradiation (CRT) to decrease toxicity and ongoing prospective clinical trials that seek to finally define the role of de-escalation of therapy for early-stage SCCA.

## 2. Chemoradiation versus Radiation Therapy Alone

Organ preservation therapy for SCCA was borne out of the pioneering work of Norman Nigro, who first used preoperative chemotherapy and radiation therapy (RT) to convert unresectable patients to resectable and incidentally found high response rates at the time of surgery, prompting him to ultimately forego resection [11,12]. Initial reports showed favorable outcomes with both RT alone as well as CRT. However, CRT was secured as the standard of care treatment for localized SCCA following two key trials, namely the United Kingdom Coordinating Committee on Cancer Research (UKCCCR) Anal Cancer Trial (ACT I) and the European Organization for Research and Treatment of Cancer (EORTC) trial [13,14,15]. In ACT I, patients treated with CRT had a reduced risk of local failure and death from anal cancer compared to the RT alone arm [13]. On long-term follow-up, CRT remained associated with a reduction in the risk of locoregional relapse, improved relapse-free and colostomy-free survival (CFS), and a reduction in the risk of dying from SCCA [15]. The outcomes of the EORTC trial were similar, with CRT resulting in improved locoregional recurrence (LRR) rates and colostomy-free interval compared to RT alone [14]. In ACT I, more acute toxicity was observed in the CRT group, with similar rates of late toxicity between the two arms [13]. While nearly 40% of patients randomized on ACT I had T1–2 N0 disease [16], patients with similar early-stage disease were excluded from the EORTC trial [14] making decisive treatment decisions challenging for this patient population. It is important to highlight that patients included in ACT I underwent essentially clinical staging alone, potentially limiting any conclusions about treatment of early-stage patients included in this trial compared to contemporary treatment that incorporates modern imaging.

Given the limited number of patients with early-stage disease included in the randomized trials that defined CRT as the standard of care for localized SCCA, questions remain as to whether RT alone is sufficient treatment for the smaller proportion of patients with early-stage disease. A summary of select studies reporting on RT alone for early-stage SCCA is shown in Table 1. In a subset analysis of patients with T1–2 N0 disease included on ACT I, a clear advantage in treatment of these patients with CRT compared to RT alone was observed for local failure (RR = 0.49, 95% CI 0.29–0.71, *p* = 0.0005) [16]. However, multiple retrospective series have shown favorable results using RT alone, particularly in those with early-stage disease [17,18,19]. A small series of 69 patients from 17 French institutions evaluated outcomes of patients with either Tis or T1 SCCA with tumor size ≤1 cm found on clinical exam or endosonography treated with RT or local excision (LE) alone [20]. Of the 69 patients included in the study, three patients with Tis underwent LE alone, while the remaining 66 were treated with RT, with 26 of those patients undergoing LE before RT. Of the 66 patients who received RT, eight underwent brachytherapy alone. The 5-year OS, CFS, and DFS rates were 94%, 85%, and 89%, respectively. 

Several smaller retrospective studies have also reported on outcomes of CRT vs. RT in patients with early-stage SCCA. Fallai et al. reported on 62 elderly (age ≥70 years) patients treated with RT or CRT for clinically staged SCCA [21]. Only 15% of patients included in the study were stage I, while 47% were stage II and 39% had stage III disease. Eight of the nine patients with stage I disease were treated with CRT with 5-year OS and locoregional control (LRC) rates of 100%. Overall, use of CRT was associated with improved 3-year outcomes, including DFS (85% vs. 46%, *p* = 0.013), local control (LC) (80% vs. 60%, *p* = 0.032), and LRC (81% vs. 61%, *p* = 0.037), but not OS (85% vs. 67%, *p* = 0.3). A review of 146 patients with T1–2 N0 SCCA treated at Geneva University Hospital in Switzerland was conducted to study the impact of concurrent chemotherapy on LRC and cancer-specific survival (CSS) [22]. Staging was completed per the treatment period, with physical examination, abdominal ultrasound, and chest radiography used initially and incorporation of abdominopelvic computed tomography (CT), transrectal echoendoscopy, pelvic magnetic resonance imaging (MRI), and fluorodeoxyglucose positron emission tomography (PET) scans for patients treated more recently. Of the patients included in the study, 80% were T2 N0 and 48% were treated with RT alone, with 91% receiving split course treatment and over half of patients receiving a brachytherapy boost to the primary tumor. The mean ± SD tumor size in the RT and CRT groups was 2.9 ± 1.2 cm and 3.2 ± 0.9 cm, respectively, *p* = 0.168. The 5-year rate of LRC for RT alone was 75.5% compared to 86.8% for patients treated with CRT, *p* = 0.155. On multivariate analysis, treatment with CRT showed a trend toward significance for LRC (HR = 2.23, 95% CI 0.95–5.23, *p* = 0.065). De Bari et al. reported on 122 patients with T1–2 N0 SCCA treated with CRT (70 patients) or RT alone (52 patients) [23]. Similar to Zilli et al., imaging for staging was dependent on the treatment time period, with only 16% of patients undergoing staging pelvic MRI. Of note, only 29% of patients treated received prophylactic inguinal irradiation, and a brachytherapy boost to the primary tumor was delivered in 68% of patients. Delivery of CRT statistically improved LC (RR = 0.34, 95% CI 0.16–0.75, *p* = 0.007).

The questionable benefit of CRT over RT alone for early-stage SCCA has also been investigated using large database studies. Buckstein et al. performed a cohort analysis comparing CRT to RT in elderly patients with stage I SCCA using the Surveillance, Epidemiology, and End Results (SEER) registry linked to Medicare [3]. The final study population consisted of 99 patients treated with RT alone and 200 patients treated with CRT. Unadjusted analysis showed that patients treated with RT alone had inferior OS at 5 years compared to those receiving CRT (61% vs. 73%, *p* = 0.002), but no difference in cause-specific survival, DFS, or rate of abdominoperineal resection was observed. Following propensity score matching, there was no significant difference in OS (*p* = 0.08) and no significant difference in the other oncologic outcomes assessed. In the matched population, CRT was associated with a higher frequency of acute toxicity but no difference in late toxicity. We sought to determine if RT alone was sufficient for treatment of patients with stage I SCCA using the National Cancer Database (NCDB) [4]. We identified 3552 stage I SCCA patients treated with CRT and 287 treated with RT alone. Patients treated with CRT were more likely to be ≥70 years old (33.1% vs. 19.7%, *p* < 0.001) and less likely to be female (63.1% vs. 71.0%, *p* < 0.001). Following propensity score matching, treatment with CRT compared to RT alone was associated with a 31% reduction in the risk of death (HR = 0.69, 95% CI 0.50–0.95, *p* = 0.023).

Talwar et al. performed a systematic review and meta-analysis comparing RT to CRT for stage I SCCA patients [5]. The authors included five retrospective studies with 3784 patients treated with CRT and 415 patients treated with RT. The 5-year OS was significantly higher for patients treated with CRT compared to RT alone (RR *=* 1.18, 95% CI 1.10–1.26, *p* < 0.00001). No difference in DFS between the groups was observed, although that endpoint was not reported in all of the studies included in the meta-analysis. Finally, of the studies able to report on toxicity, higher rates of both acute and late toxicity were reported in patients who received CRT.

## 3. Local Excision versus Chemoradiation

While the standard of care for localized SCCA is CRT, there has been some debate as to whether CRT is the optimal approach for patients with early-stage SCCA, particularly T1 N0 disease. This is because patients with T1 N0 disease represented a very small proportion of patients treated on the major randomized studies of SCCA [13,14,24,25,26]. The number of series that report LC results of local excision alone are few, all with a small number of patients, some including T1 and T2 tumors, and all demonstrating local recurrence rates ranging 0–60% [27,28,29,30,31]. In a recent retrospective study of 57 patients with T1 N0 SCCA, 13 received local excision alone and 44 received CRT [32]. Local recurrences occurred in two of the 13 patients treated with LE (15% rate, both salvaged with surgery) and one of the 44 patients with CRT. There was no difference in 5-year progression-free survival (PFS) between the two cohorts (91% vs. 83%, *p =* 0.57). A large retrospective study of patients with T1 N0 SCCA identified from the NCDB compared OS amongst 503 patients treated with local excision alone compared to 1740 patients treated with CRT from 2004 to 2012 [33]. The authors found that the use of local excision alone increased steadily during the study period from a rate of 17.3% in 2004 to 30.8% in 2012. The 5-year OS rate was similar in patients treated with local excision alone compared to CRT (85.3% vs. 86.8%, *p =* 0.93). Interestingly, a comprehensive NCDB analysis of treatment outcomes in patients with stage I SCCA from 2004 to 2015 found slightly worse OS in patients treated with excision alone compared to CRT in a propensity-score-matched cohort (4-year OS 82.8% vs. 85.6%, *p =* 0.045) [4]. While these OS data from the NCDB studies are somewhat conflicting in a statistical manner, the OS rates of 83–85% for excision alone compared to 86–87% are numerically similar and likely not clinically significant differences. Therefore, excision alone can be considered for select patients with T1 N0 disease, which is supported by the NCCN guidelines [34].

## 4. Modifications to Systemic Therapy

Radiation therapy with concurrent 5-FU and mitomycin C (MMC) remains the current standard of care for localized SCCA based on the results of multiple large randomized trials. The addition of MMC to 5-FU and RT was investigated in RTOG 87-04/ECOG 1289 [25]. Adding MMC improved the 4-year local failure rate (34% vs. 16%, *p =* 0.0008), DFS (51% vs. 73%, *p =* 0.0003), and CFS (59% vs. 71%, *p =* 0.014) with no significant difference in OS (67% vs. 76%, *p =* 0.31) but at the cost of increased grade 4–5 toxicity (8% vs. 26%, *p* < 0.001). In an attempt to replace MMC, several studies compared replacing MMC for cisplatin with concurrent 5-FU and RT [24,26,35]. RTOG 98-11 was a phase III randomized trial comparing 5-FU plus cisplatin induction chemotherapy followed by RT with concurrent 5-FU plus cisplatin to RT with concurrent 5-FU plus MMC [24]. The initial results showed an improvement in colostomy rate with the use of MMC (10% with MMC vs. 19% with cisplatin, *p =* 0.02), but no difference in 5-year LR, DFS, or OS despite higher rates of severe acute grade 3–4 hematologic toxicity with MMC (61% with MMC vs. 42% with cisplatin, *p* < 0.001). At longer follow-up, DFS (67.8% vs. 57.8%, *p =* 0.006) and OS (78.3% vs. 70.7%, *p =* 0.026) at 5 years were statistically better for RT with concurrent 5-FU plus MMC compared to 5-FU plus cisplatin [35]. In addition, a trend for improved CFS (71.9% vs. 65.0%, *p =* 0.05), locoregional failure (20.0% vs. 26.4%, *p =* 0.087), and colostomy failure (11.9% vs. 17.3%, *p =* 0.074) with the addition of MMC was also reported. ACT II was a 2 × 2 factorial trial that investigated if replacing MMC with cisplatin improves response and if maintenance chemotherapy following CRT improves PFS [26]. Patients were randomized to receive either MMC or cisplatin with concurrent 5-FU and RT with or without two additional courses of 5-FU and cisplatin. No difference in 3-year PFS was observed between the MMC and cisplatin groups (HR *=* 0.95, 95% CI 0.75–1.19, *p =* 0.63) or in those patients receiving maintenance chemotherapy. Of note, similar toxicity was observed in the MMC and cisplatin groups.

Modifications to this standard regimen have resulted in similar rates of efficacy with potential improvements in toxicity. The replacement of 5-FU by capecitabine has been investigated in multiple smaller phase II trials and retrospective reports. A multicenter phase II pilot study was performed in the United Kingdom that investigated the use of capecitabine delivered at a dose of 825 mg/m^2^ twice daily during RT instead of 5-FU using the ACT II radiation regimen (50.4 Gy with concurrent MMC 12 mg/m^2^ delivered on day 1) [36]. The LC rate based on exam, imaging, or both at 6 months following completion of treatment was 90%. When evaluating compliance with treatment, 58% of patients completed both chemotherapy and RT as planned. Grade 3–4 acute toxicity was observed in 45% of patients. Oliveira et al. conducted a phase II trial consisting of 43 patient with either T2–4 N0 M0 or Tany N1–3 M0 SCCA treated with capecitabine 825 mg/m^2^ twice daily during RT with a single dose of MMC 15 mg/m^2^ on day 1 [37]. The primary endpoint of the study was LC at 6 months determined by clinical exam and imaging with either pelvic CT or MRI. The rate of LRC at 6 months was 86%. The main grade 3–4 toxicities were grade 3 radiation dermatitis (23.2%), grade 3 lymphopenia (11.6%), and grade 3 neutropenia (6.9%). Treatment interruption of capecitabine was required in 55.8% of patients for a mean duration of 11.2 ± 11.0 days due to primarily grade 3 radiation dermatitis and grade 2–3 hematologic toxicity. 

Treatment in a larger cohort of patients receiving more modern treatment was conducted by Jones et al., who reviewed SCCA patients treated with intensity-modulated radiation therapy (IMRT) with concurrent MMC (single-dose on day 1) and either 5-FU or capecitabine from 50 centers in the United Kingdom [38]. Data from 40 centers with 147 patients were included in the study, 35.4% of whom were treated with concurrent capecitabine/MMC and 64.6% treated with concurrent 5-FU/MMC. Although limited by available data, the 1-year relapse-free rates were not significantly different between the two groups (76.2% in the capecitabine/MMC vs. 79.3% in the 5-FU/MMC, *p =* 0.80), nor were the 1-year CFS rates (77.5% in the capecitabine/MMC vs. 90.7% in the 5-FU/MMC, *p =* 0.09). While no difference was observed in overall rates of grade 3 or 4 toxicity (45% for capecitabine/MMC vs. 55% for 5-FU/MMC), less grade 3–4 hematologic toxicity was observed in the capecitabine/MMC cohort (4% vs. 27%, *p* < 0.001). A single institution series by Goodman et al. compared 107 patients with SCCA treated with IMRT and concurrent MMC (day 1 and 29 at 10 mg/m^2^) with 5-FU (63 patients) or capecitabine (44 patients) [39]. The 2-year oncologic outcomes between the groups were similar, including OS (87% for 5-FU vs. 98% for capecitabine, *p =* 0.12), LRR (6.5% for 5-FU vs. 8.2% for capecitabine, *p =* 0.78), distant metastasis (14.7% for 5-FU vs. 7.6% for capecitabine, *p =* 0.26), and colostomy rate (5% for 5-FU vs. 9% for capecitabine, *p =* 0.65). More grade 3 to 4 neutropenia was observed in the 5-FU group compared to the capecitabine group (52% vs. 20%, *p =* 0.001). Treatment breaks due to toxicity were more likely in the 5-FU group compared to the capecitabine group (41% vs. 14%, *p =* 0.006) with a median treatment duration significantly longer for patients receiving 5-FU (39 days, range 32–52 days vs. 37 days, range 32–44 days, *p* < 0.001). Finally, dose reductions were more likely in the 5-FU group (52% vs. 16%, *p* < 0.001). A meta-analysis of five trials reported pooled outcomes, including a complete response rate of 88% (83–94%) at 6 months post-treatment with an overall complete response rate evaluated at different time intervals of 91% (87–95%), with 93.5–100% of patients completing the planned RT dose [40]. This result further highlights that capecitabine is an effective and convenient alternative to 5-FU for treatment of SCCA. 

Dosing of MMC has varied between North American and European trials. While in North American trials, MMC has been dosed as 10 mg/m^2^ delivered in two doses, European trials typically deliver 12–15 mg/m^2^ on day 1 of RT [13,14,24,25,26]. Reducing the dose of MMC conceivably would reduce toxicity. White et al. performed a single-institution retrospective review of 217 patients receiving definitive CRT for localized SCCA, comparing treatment outcomes and toxicity in patients who received one (154 patients) vs. two (63 patients) cycles of concurrent MMC [41]. At 2-years, no significant differences in oncologic outcomes were observed between the two groups (one vs. two cycles), including PFS (78% vs. 85%, *p =* 0.39), CSS (88% vs. 94%, *p =* 0.11), CFS (87% vs. 92%, *p =* 0.51), and OS (84% vs. 91%, *p =* 0.16). Rates of grade ≥2 acute toxicity were higher in the two cycle group, including overall hematologic (89% vs. 73%, *p =* 0.01), skin (97% vs. 84%, *p =* 0.006), genitourinary (19% vs. 8%, *p =* 0.04), and treatment-related death (5% vs. 0%, *p =* 0.02). While limited, the results of this study indicate that a single dose of MMC is efficacious and may result in less acute treatment-related toxicity than two cycles. Of course, this result needs to be confirmed in prospective trials.

## 5. Advancements in Radiation Therapy

### 5.1. Intensity-Modulated Radiation Therapy (IMRT)

While CRT is the standard of care for management of localized SCCA, acute toxicities of this therapy represent a major challenge for patients. Until the early 2000s, most patients were treated with conventional RT, which refers to the use of either two-dimensional (2D) or three-dimensional (3D) conformal radiation therapy (3DCRT) techniques. The field design for 2D/3DCRT generally included an anterior–posterior (AP) field and a posterior–anterior (PA) field, where the whole pelvis (superior border at L5/S1) was treated to 30.6 Gy. At that point, the superior border of the AP/PA fields was then reduced to the bottom of the sacroiliac joints to a total dose of 45 Gy. The fields would be reduced again, and the primary tumor would then receive an additional boost dose of radiation. The inguinal nodes were generally either included in the AP field by widening the field size (wide AP/narrow PA; “thunderbird technique”), or the inguinal nodes were supplemented using electrons [42,43]. While RT is intended to target the primary tumor and regional lymphatics, the conventional 2D/3DCRT techniques described above result in the irradiation of many other organs and normal tissues (small bowel, large bowel, bladder, bone marrow, genitalia, and external skin). The acute toxicities that subsequently develop from irradiation of these nontarget structures may result in radiation treatment interruptions, thereby prolonging the overall treatment time. Rates of grade 3 or higher nonhematologic toxicities are as high as 61% [13,14,35,44]. These severe acute toxicities include dermatitis, diarrhea, dysuria, and proctitis. Furthermore, grade 3 or higher hematologic toxicity rates are as high as 60% in studies that have incorporated MMC [25,35,45]. In addition to these severe acute effects of concurrent chemoradiation, long-term toxicities, such as anal stenosis, sphincter dysfunction, sexual dysfunction, and fibrosis, are common long-term effects of CRT for anal cancer. In a systematic review of the literature, Pan et al. found that the overall incidence of late gastrointestinal toxicity in over 130 studies of anal cancer ranged from 7% to 64.5%, with grade 3 and higher toxicities reported in up to 33.3% of patients [46]. The most common late toxicities were fecal incontinence (up to 44%), diarrhea (up to 27%), and ulceration (up to 23%) [46]. These severe and common acute and long-term effects of CRT for patients with SCCA underscores the need for improved radiation techniques as well as the need for de-escalation of radiation dose in appropriate situations.

Approximately 15 years ago, the first reports of IMRT for the treatment of SCCA began to emerge [47,48]. IMRT uses an inverse-planning algorithm that varies beam intensities from multiple different angles to allow the prescription dose to conform tightly around the target volumes while decreasing radiation dose to surrounding normal tissues. IMRT can be delivered with static, fixed fields or with rotational therapy (i.e., volumetric modulated arc therapy, VMAT). Early proof of principle studies demonstrated that IMRT can reduce radiation dose to surrounding structures, such as the bladder, bowel, and perineal skin, with resulting acceptable toxicity and clinical outcome profile [47,48]. There have been no randomized studies of IMRT compared to conventional radiation therapy techniques, but several retrospective studies have compared the clinical outcomes and toxicities in patients with SCCA treated with IMRT versus 2D/3DCRT [42,49,50]. The first study to compare these approaches demonstrated that, compared to patients treated with 2D/3DCRT (*N =* 17), patients treated with IMRT (*N =* 29) had less acute grade >2 nonhematologic toxicities (65% 2D/3DCRT vs. 21% IMRT, *p =* 0.003), reduced need for treatment breaks (88% 2D/3DCRT vs. 34.5% IMRT, *p =* 0.001), and shorter treatment duration (57 days 2D/3DCRT vs. 40 days IMRT, *p* < 0.0001) [42]. One of the early concerns with IMRT was that the more conformal dose to the target volumes and steep dose gradient might actually result in an increased risk of LRR. However, early results of patients treated with IMRT in these retrospective studies showed 2–3 year local–regional control rates of 87–91% compared to 57–87% with 2D/3DCRT [42,49]. Finally, in a large, retrospective study (*N =* 376 IMRT; *N =* 403 2D/3DCRT) utilizing the Veterans Affairs database, Bryant et al. demonstrated that patients treated with IMRT had significantly lower rates of treatment breaks ≥5 days, increased rates of receiving both doses of MMC, and decreased risk of colostomy related to tumor recurrence or progression [50]. These authors found no difference in CSS in patients treated with IMRT vs. 2D/3DCRT.

Prospective evaluation of IMRT for SCCA has been conducted in the RTOG 0529 clinical trial [45]. In this study, dose-painted IMRT (DP-IMRT) was used to treat elective nodal volumes to a lower dose (42 Gy in 28 fractions for T2 N0 disease and 45 Gy in 30 fractions for T3–4 N0–3 disease) while simultaneously treating the gross disease to a higher dose (50.4 Gy in 28 fractions to the primary tumor for T2 N0 disease, 54 Gy in 30 fractions to the primary tumor for T3–4 N0–3 disease, 50.4 Gy in 30 fractions for gross nodal disease ≤3 cm, and 54 Gy in 30 fractions for gross nodal disease >3 cm). Compared to the control arm from the RTOG 98-11 study (5-FU/MMC/RT), there was a significant reduction in acute grade 2+ hematologic toxicity (73% vs. 85%, *p =* 0.03), grade 3+ gastrointestinal toxicity (21% vs. 36%, *p =* 0.0082), and grade 3+ dermatologic toxicity (23% vs. 49%, *p* < 0.0001) with DP-IMRT [45]. Long-term cancer control outcomes from this study have not yet been published.

With respect to acute hematologic toxicities, there have been a series of studies over the past 10–15 years investigating the association of radiation dose to the pelvic bone marrow (PBM) and the subsequent development of acute hematologic toxicities. These early studies demonstrated that low dose to the pelvic bone marrow (5–20 Gy) is associated with endpoints such as grade ≥2 hematologic toxicities [51,52,53]. Subsequent studies have used advanced imaging modalities now used in the staging of SCCA, including PET/CT, to identify the active portions of the bone marrow to help further facilitate bone marrow sparing [54,55,56]. There are now prospective data to support the use of IMRT for bone marrow sparing in an effort to reduce acute severe hematologic toxicities [57]. 

Recently, a couple of retrospective studies reported long-term results of treating patients with SCCA with IMRT. Investigators from the Mayo Clinic reported results of 127 patients with SCCA treated from 2003 to 2019 with IMRT with a median follow-up of 4 years [58]. The 4-year LRR was 9%. Acute grade 3+ toxicity rates were 31% hematologic, 17% gastrointestinal, and 16% dermatologic, comparing favorably with the RTOG 0529 study. Grade 3+ long-term effects were uncommon and included 3% gastrointestinal toxicity, 2% genitourinary toxicity, and 1% pain. With a median follow-up of nearly 6 years, de Meric de Bellefon et al. reported long-term results of 193 patients with SCCA treated with IMRT [59]. In a heterogeneous cohort that included 63% of patients with stage III disease, the long-term LC was excellent at 89%. With longer follow-up, these authors did find that there was a 24% rate of grade 3 late toxicities (the most common included vaginal stricture/pain with intercourse/discharge or rectovaginal fistula in 27 patients, proctitis or rectal ulcer in 14 patients, and diarrhea in 12 patients) and one patient with late grade 4 hematuria.

Taken together, these data suggest that radiation therapy with IMRT should be a standard of care for patients with localized SCCA. There is substantial evidence that has accumulated to demonstrate that acute toxicities and treatment breaks/delays are significantly lower in patients treated with IMRT compared to conventional radiation therapy techniques. In addition, more evidence has accumulated that IMRT results in at least the same cancer control outcomes compared with conventional techniques. 

### 5.2. Adjusting Radiation Therapy Volumes 

IMRT represents an avenue of radiation de-escalation by reducing the dose to surrounding critical structures. Another avenue of interest in radiation de-escalation has been omission of RT to the inguinal nodal basins in patients that present with uninvolved inguinal nodes. Irradiation of the inguinal nodes increases the risk of acute dermatologic toxicity and may also be associated with long-term complications, such as lower extremity lymphedema. However, inguinal nodal irradiation has been a component of early randomized clinical trials [24,25,26], though it was systematically omitted in one study [14] and left to the physicians’ choice in another [60]. In order to help answer the question regarding necessity of elective inguinal irradiation, Ortholan et al. conducted a retrospective, multicenter study involving four cancer centers in south France that treated SCCA between 2000 and 2004 [61]. Patients were staged by clinical exam, ultrasound, endorectal ultrasonography, and CT without use of PET. Amongst 181 patients with uninvolved inguinal nodes, the decision to treat the inguinal lymph nodes was per the discretion of the physician: 75 received elective inguinal irradiation (45 Gy in 25 fractions) and 106 did not. The groups were well balanced with the exception that those that received inguinal irradiation tended to have larger tumors and be of younger age. The cumulative rate of inguinal recurrence was 2% in those that received irradiation vs. 16% in those that did not. When analyzed by T stage, those that did not receive inguinal irradiation had >10% risk of inguinal nodal recurrence (12% for T1–2 tumors and 30% for T3–T4 tumors).

There are some contrasting data to the Ortholan et al. study. Crowley et al. reported results of 30 patients with SCCA, none of whom had inguinal nodal or pelvic nodal involvement based on clinical exam and cross-sectional imaging, treated with CRT at a single center from 1998 to 2004 [62]. The radiation fields did not include the inguinal nodes in any cases, and all patients had T1–3 N0 disease. At a median follow-up of 41 months, there was only one patient with an inguinal nodal relapse, suggesting that omission of inguinal nodal RT may be safe. Similarly, in a study from the Samsung Medical Center in Korea, Kim et al. retrospectively reviewed 33 patients treated with CRT for SCCA between 1994 and 2013, all of whom had no inguinal nodal involvement at diagnosis based on clinical exam and CT scans in the majority of patients and pelvic MRI (42.4%) and PET (21.2%) in the minority [63]. None of these patients received elective inguinal nodal irradiation. At a median follow-up of 50 months, the authors found no cases of inguinal nodal recurrence.

While the data from these two small studies are compelling, the data from Ortholan et al. strongly suggest that all patients with T3–4 disease should receive inguinal irradiation. At this time, there is insufficient data to suggest that omission of inguinal nodal irradiation is safe even for T1–2 tumors, especially in light of the 12% risk of inguinal nodal relapse seen in these patients in the Ortholan et al. study. However, in the modern era with excellent staging techniques, including use of PET/CT, prospective evaluation of omission of inguinal nodal RT or lower doses of inguinal nodal RT is worthy of study in low-risk patients (T1–2 N0).

### 5.3. Adjusting Radiation Therapy Dose

Finally, another method of radiation de-escalation is to lower the total RT dose to some or all of the target volumes. In the original Nigro protocol [12], the RT dose described was 3000 cGy in 15 fractions to the full pelvis along with 5-FU/MMC and was intended to be given as preoperative therapy prior to APR. In all three of the patients initially described, the tumor had a complete clinical response. Subsequent updates in larger patient numbers continued this approach of 3000 cGy in 15 fractions with 5-FU/MMC and showed excellent LC rates [11,64]. The approach of lowering the total dose may be adequate for smaller tumors. For instance, Smith et al. reported results of 42 consecutive patients treated with 30 Gy/15 fractions along with 5-FU/MMC [65]. The LC was 90% for patients with T1–2 tumors but only 38% for patients with T3–4 disease. In a separate series of 21 patients with T1 N0 disease based on clinical exam and CT treated with 30 Gy/15 fractions + 5-FU/MMC, Hatfield et al. found that only one patient experienced a local relapse at a median follow-up of 42 months [66]. In addition to treating these patients with reduced-dose RT, 18 of the 21 patients also had treatment fields that were of smaller volume encompassing the gross tumor with a 3 cm margin in all directions as opposed to larger fields that encompassed the whole pelvis. The approach of reduced-dose RT has also been examined by Charnley et al. in a group of elderly patients (>75 years old) with poor performance status precluding the use of standard-dose CRT [67]. In this retrospective study, 16 patients (10 T2 N0 and 6 T3–4 N0–3; 81% completed CT scans for staging) received 30 Gy in 15 fractions to fields encompassing the gross tumor + 3 cm in all directions along with 5-FU (no MMC). Median follow-up was 16 months, and the LC rate was 73%.

In summary, there are some data for reduced-dose RT. Reducing RT dose appears to be a promising treatment strategy for patients with small, node-negative tumors (e.g., T1–2 N0) and is now the subject of at least two prospective trials. The optimal treatment approach for elderly patients and/or those with poor PS that may preclude standard-dose RT and chemotherapy regimens also remains an area worthy of future investigation.

## 6. Current Prospective Trials

De-escalation of therapy for early-stage SCCA is currently being investigated in multiple prospective clinical trials as summarized in Table 2. DECREASE is a clinical trial evaluating lower-dose CRT in early-stage SCCA (T1–2 (with tumor size ≤4 cm) N0 M0) currently accruing through the Eastern Cooperative Oncology Group (NCT04166318). Patients complete staging studies with specific criteria to assess risk of cancer involvement based on imaging modality and are then randomized to standard-dose CRT or deintensified CRT. In standard-dose CRT, the primary tumor receives 50.4 Gy, and the elective nodal regions, including the full pelvis and inguinal nodes, receive a dose of 42 Gy in 28 fractions with concurrent MMC (single dose of 12 mg/m^2^) and two cycles of 5-FU or concurrent capecitabine. With deintensified CRT, the primary tumor and pelvis/inguinals receive a dose of 36 and 32 Gy in 20 fractions, respectively, for T1 N0 disease and 41.4 and 34.5 Gy in 23 fractions, respectively, for T2 N0 disease with concurrent MMC (single dose of 10 mg/m^2^) and one cycle of 5-FU or capecitabine. The coprimary endpoint of the study is to determine if deintensified CRT can maintain 2-year disease control of 85% or higher while improving anorectal health-related quality of life compared to standard-dose CRT.

The PLATO (PersonaLising Anal cancer radioTherapy dOse) umbrella trial (ISRCTN88455282) is being conducted by Cancer Research UK and includes clinical trials ACT 3 and 4. ACT 3 is a nonrandomized phase II study evaluating local excision with selective postoperative CRT for patients with T1 N0 anal margin tumors. Patients with surgical margins >1 mm will receive no additional treatment, while those with margins ≤1 mm receive additional CRT with reduced doses (41.4 Gy in 23 fractions with single dose MMC and concurrent capecitabine) [68]. ACT 4 is a randomized phase II trial comparing reduced-dose (41.4 Gy in 23 fractions) to standard-dose (50.4 Gy in 28 fractions) CRT for patients with T1–2 (<4 cm) N0 SCCA with the goal of decreasing toxicity while maintaining high rates of LRC [69].

The dosimetric advantages of proton therapy may also be a useful method to reduce toxicity and spare adjacent organs at risk in the management of SCCA. Use of proton therapy to treat localized SCCA while sparing the pelvic kidney in transplant recipients has been reported in a small case series [70]. In a multi-institutional single-arm pilot study, patients with localized SCCA were treated with pencil beam scanning proton beam RT with concurrent 5-FU and MMC [71]. Of the 25 patients enrolled in the protocol, 23 completed treatment per protocol. The primary endpoint was feasibility of combination treatment with a grade 3+ dermatologic toxicity rate less than 48% (from RTOG 98–11). The grade 3+ radiation dermatitis rate was 24% with an overall clinical complete response rate of 88%. There are two ongoing trials investigating the use of proton beam therapy for SCCA. One study is a pilot study in patients with locally advanced disease evaluating the feasibility of intensity-modulated proton beam therapy with concurrent 5-FU and MMC to reduce toxicity in SCCA, being conducted at the University of Cincinnati (NCT03018418). The primary endpoint of the study is to evaluate rates of grade 3+ acute hematologic, gastrointestinal, genitourinary, and dermatologic toxicity. The second study is a phase II feasibility trial at MD Anderson Cancer Center (NCT03690921) investigating the use of linear energy transfer-optimized intensity-modulated proton therapy for definitive chemoradiation (concurrent cisplatin and 5-FU) of stages I–III SCCA. The primary endpoint is physician-reported grade 3+ gastrointestinal, genitourinary, and hematologic toxicity. If the results of such studies are promising, proton beam therapy may be another method to further reduce toxicity in the treatment of patients with early-stage SCCA.

## 7. Conclusions

Determining appropriate de-escalation of therapy for early-stage SCCA is of utmost importance given the aging population at risk of this malignancy and is an active area of clinical investigation, with multiple currently enrolling critical clinical trials. While the results of these trials will start to shed light on the appropriate use of reducing intensity of treatment in the general population of early-stage patients, additional questions remain regarding the optimal treatment of the elderly or those with a less favorable performance status. Perhaps future studies will take advantage of advances in radiation delivery, such as proton beam therapy and novel biomarkers, to further customize therapy and in turn reduce toxicity in these favorable prognosis early-stage SCCA patients.

## Figures and Tables

**Table 1 cancers-13-02099-t001:** Select studies evaluating radiation therapy (RT) alone or comparing chemoradiation (CRT) to RT in early-stage anal cancer.

Author	Inclusion	Key Results
Ortholan et al. [20]	69 patients: 12 patients with Tis, 57 patients with T1, all ≤1 cm; 66 received RT, 3 Tis treated with local excision alone	91% local control in RT group; 5-year OS 94%, CFS 85%, and DFS 89%
Fallai et al. [21]	62 patients: 9 stage I, with 8 patients treated with CRT	5-year OS and LRC both 100% for stage I patients
Zilli et al. [22]	146 patients: 29 patients with T1, 117 with T2 disease; RT alone in 71 and CRT in 75	5-year LRC of 75.5% for RT vs. 86.8% for CRT, *p =* 0.155; 5-year CSS of 88.5% for RT vs. 94.9% for CRT, *p =* 0.161
De Bari et al. [23]	122 patients: 24 patients with T1, 98 patients with T2; RT alone in 52 and CRT in 70	CRT improved LC on multivariate analysis (RR *=* 0.34, 95% CI 0.16–0.75, *p =* 0.007)
Miller et al. [4]	3839 stage I patients: RT alone in 287 and CRT in 3552	CRT associated with a 31% reduction in the risk of death compared to RT alone (HR *=* 0.69, 95% CI 0.50–0.95, *p =* 0.023)
Buckstein et al. [3]	299 stage I patients: RT alone in 99 and CRT in 200	After propensity score matching, no difference in OS, CSS, CFS, or DFS between the groups.

**Table 2 cancers-13-02099-t002:** Summary of current de-escalation studies for early-stage squamous cell carcinoma of the anus.

Study	Key Inclusion Criteria	Study Design/Treatment	Primary Endpoint
DECREASE (NCT04166318)	T1–2 (with tumor size ≤4 cm) N0 anal canal or anal margin squamous cell carcinoma; specific radiographic criteria for lymph node evaluation.	Randomized phase II in 1:2 fashion to standard-dose CRT vs. deintensified CRT. Doses of RT based on T stage.	To determine if deintensified CRT results in 2-year disease control ≥85% while improving health-related quality of life compared to standard CRT.
ACT 3 (ISRCTN88455282)	T1 N0 anal margin squamous cell carcinoma treated with local excision.	Nonrandomized phase II: patients with tumor margins >1 mm will undergo observation, while those with margins ≤1 mm will receive adjuvant CRT.	To assess the 3-year locoregional failure rate.
ACT 4 (ISRCTN88455282)	T1–2 (with tumor size ≤4 cm) N0 anal canal or T2 (with tumor size ≤4 cm) N0 anal margin squamous cell carcinoma.	Randomized phase II in 1:2 fashion to standard-dose CRT vs. deintensified CRT.	To assess the 3-year locoregional failure rate.

## Data Availability

The data presented in this study are available in this article.

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
