# Peer review of "De-Escalation of Therapy for Patients with Early-Stage Squamous Cell Carcinoma of the Anus"

_cancers, 2021, doi:10.3390/cancers13092099_

Round 1
Reviewer 1 Report
this is a well written organised comprehensive review of the available literature.
the crucial missing point in terms of interrogation of the date is the staging undertaken to derive the apparent stage.
In particular early studies of which ACT 1 is a good example has affectively clinical staging only. I never believed that 40% of cases in this study were early stage disease judging by our own experience. MRI and PET scans have only been more widely used in the last probably 10 years and not necessarily in UK practice until the most recent studies.
it is therefore entirely plausible that much of the evidence driven need to give chemo-radiation for apparent early stage disease is simply an effect of under staging and thus under reporting the disease treated.
I therefore think it important to have a critical analysis of the staging undertaken of each of the studies that you report to give a degree of context. In my opinion this is likely to be crucial in the current studies that are underway in determining whether deescalation is safe or not
Author Response
Reviewer 1:
this is a well written organised comprehensive review of the available literature.
the crucial missing point in terms of interrogation of the date is the staging undertaken to derive the apparent stage.
In particular early studies of which ACT 1 is a good example has affectively clinical staging only. I never believed that 40% of cases in this study were early stage disease judging by our own experience. MRI and PET scans have only been more widely used in the last probably 10 years and not necessarily in UK practice until the most recent studies.
it is therefore entirely plausible that much of the evidence driven need to give chemo-radiation for apparent early stage disease is simply an effect of under staging and thus under reporting the disease treated.
I therefore think it important to have a critical analysis of the staging undertaken of each of the studies that you report to give a degree of context. In my opinion this is likely to be crucial in the current studies that are underway in determining whether de-escalation is safe or not
Our response: Thank you for these helpful comments. We agree with the reviewer’s thoughts and have added additional text to the introduction mentioning this as a limitation to prior studies (47-50) as well as each of the studies mentioned in the manuscript including:
ACT I – lines 75-78
Ortholan et al. – line 89
Fallai et al. – line 97
Zilli et al. – lines 105-109
De Bari et al. – lines 117-119
Ortholan et al. – lines 373-374
Crowley et al. – line 385
Kim et al. – lines 391-392
Hatfield et al. – line 414
Charnley et al. – line 422
Additional text was also added to the discussion of the DECREASE trial to highlight the importance of staging imaging in determining those with node negative disease (lines 436-437).
Reviewer 2 Report
The manuscript gives an overview of current strategies for treatment deintensification in anal squamous cell carcinoma. With the excellent in prognosis in T1-2 N0 patients this is a topic of great importance. The manuscript is well written and adresses the different strategies that are currently employed. There are some minor improvements that will make this manuscript suitable for publication.
1. In the introduction the excellent prognosis for early stage patients should be more clearly elaborated on using the data from the randomized trials.
2. The different goals of deintensification e.g. reduction of late vs acute toxicities should also be discussed - especially when talking about omission of chemotherapy.
3. The reduction of radiotherapy dose should be discussed more broadly with regards to potentially less (severe) long term side effects. Fecal incontinence, procitis rates should be discussed here (https://pubmed.ncbi.nlm.nih.gov/30264638/ as an excellent literature review on this topic). These long term effects are more related to radiotherapy than chemotherapy so a dose reduction could certainly be warranted.
Author Response
Reviewer 2:
The manuscript gives an overview of current strategies for treatment deintensification in anal squamous cell carcinoma. With the excellent in prognosis in T1-2 N0 patients this is a topic of great importance. The manuscript is well written and adresses the different strategies that are currently employed. There are some minor improvements that will make this manuscript suitable for publication.
1. In the introduction the excellent prognosis for early stage patients should be more clearly elaborated on using the data from the randomized trials.
Our response: Thank you for this helpful comment. Additional text was added to the introduction (lines 44-47) highlighting the excellent prognosis of this patient group.
2. The different goals of deintensification e.g. reduction of late vs acute toxicities should also be discussed - especially when talking about omission of chemotherapy.
Our response: Thank you for this comment. We have added additional text to highlight potential opportunities to reduce acute and late toxicity associated with chemoradiation for treatment of early stage anal cancer – lines 281-294, 333-341.
3. The reduction of radiotherapy dose should be discussed more broadly with regards to potentially less (severe) long term side effects. Fecal incontinence, procitis rates should be discussed here (https://pubmed.ncbi.nlm.nih.gov/30264638/ as an excellent literature review on this topic). These long term effects are more related to radiotherapy than chemotherapy so a dose reduction could certainly be warranted.
Our response: Thank you for this helpful feedback and for the reference. We have added this reference and additional discussion regarding late toxicities associated with chemoradiation and potential for reduction of these toxicities with appropriate de-escalation strategies – lines 287-294.
Round 2
Reviewer 1 Report
Thanks for addressing my concerns